# Interactions between Trypillian farmers and North Pontic forager-pastoralists in Eneolithic central Ukraine

Alexey G. Nikitin[1]*, Mykhailo Videiko[2], Nick Patterson[3,4], Virginie Renson[5], David Reich[3,4,6,7]

1 Department of Biology, Grand Valley State University, Allendale, Michigan, United States of America, 2 Scientific Research Laboratory of Archaeology, Borys Grinchenko Kyiv University, Kyiv, Ukraine, 3 Broad Institute of Harvard and MIT, Cambridge, Massachusetts, United States of America, 4 Department of Human Evolutionary Biology, Harvard University, Cambridge, Massachusetts, United States of America, 5 University of Missouri Research Reactor, Columbia, Missouri, United States of America, 6 Department of Genetics, Harvard Medical School, Boston, Massachusetts, United States of America, 7 Howard Hughes Medical Institute, Harvard Medical School, Boston, Massachusetts, United States of America

* nikitin@gvsu.edu

**Data Availability Statement:** The aligned sequences are available through the European Nucleotide Archive under accession number PRJEB61755. Otherwise, all data generated or

## Abstract

The establishment of agrarian economy in Eneolithic East Europe is associated with the Pre-Cucuteni-Cucuteni-Trypillia complex (PCCTC). PCCTC farmers interacted with Eneolithic forager-pastoralist groups of the North Pontic steppe as PCCTC extended from the Carpathian foothills to the Dnipro Valley beginning in the late 5th millennium BCE. While the cultural interaction between the two groups is evident through the Cucuteni C pottery style that carries steppe influence, the extent of biological interactions between Trypillian farmers and the steppe remains unclear. Here we report the analysis of artefacts from the late 5th millennium Trypillian settlement at the Kolomiytsiv Yar Tract (KYT) archaeological complex in central Ukraine, focusing on a human bone fragment found in the Trypillian context at KYT. Diet stable isotope ratios obtained from the bone fragment suggest the diet of the KYT individual to be within the range of forager-pastoralists of the North Pontic area. Strontium isotope ratios of the KYT individual are consistent with having originated from contexts of the Serednii Stih (Sredny Stog) culture sites of the Middle Dnipro Valley. Genetic analysis of the KYT individual indicates ancestry derived from a proto-Yamna population such as Serednii Stih. Overall, the KYT archaeological site presents evidence of interactions between Trypillians and Eneolithic Pontic steppe inhabitants of the Serednii Stih horizon and suggests a potential for gene flow between the two groups as early as the beginning of the 4th millennium BCE.

## Introduction

Trypillian culture (5000–2750 BCE) is the eastern component of the Pre-Cucuteni-Cucuteni-Trypillia complex (PCCTC) of Eneolithic farmers of eastern Europe. PCCTC extended from the Carpathian Mountains to the Dnipro (Dnieper) River in what is now Moldova, Romania

analyzed during this study are included in this published article and its Supporting Information files.

**Funding:** This study was supported, in part, by Professional Development Funds from the College of Liberal Arts and Sciences of Grand Valley State University to AGN and by the Howard Hughes Medical Institute (HHMI) to DR. The acquisition of the Nu Plasma II MC-ICP-MS was funded by the National Science Foundation (grant #BCS-0922374). The Archaeometry Laboratory is supported by the National Science Foundation (grant #BCS-1912776). The funders had no role in study design, data collection and analysis, decision to publish, or preparation of the manuscript.

**Competing interests:** The authors have declared that no competing interests exist.

and Ukraine. PCCTC is known by over 4300 settlements and cemeteries. In the 5th millennium BCE PCCTC was synchronous with such European Neolithic complexes as Lengyel, Vinca, and Kodjadermen-Gumelnița-Karanovo VI (KGK). In the east, Trypillia, the Ukrainian branch of PCCTC, neighbored the Dnipro-Donets and Pit-Comb Ware fisher-forager groups, as well as the Serednii Stih (Sredny Stog) forager-pastoralists. The Serednii Stih archaeological horizon formed in the steppe of Azov, between the Dnipro and the Don Rivers, in the 5th millennium BCE. It existed, in various cultural forms, through the end of the 4th millennium BCE [1], until the Yamna cultural complex took over the steppe dominance. Serednii Stih and Yamna are viewed as an Eneolithic-Bronze Age cultural continuum [2] and Yamna is considered an important source for the spread of Indo-European cultural traits across a much larger geographic region beginning in the Bronze Age [3].

Close contacts between Trypillia and Serednii Stih began as early as the end of the Trypillia A period (ca. 4700–4600 BCE) and continued throughout the Eneolithic [1]. Archaeological evidence shows reciprocal influence of Serednii Stih and Trypillia on each other's cultural development. Settlements of the Deriivka group of the Serednii Stih horizon such as Deriivka II and Molyukhiv Bugor (Fig 1) contain evidence of agricultural practices, likely influenced by the neighboring Trypillian groups [1]. The influence of Serednii Stih on Trypillian culture is most evident in the presence of Serednii Stih ceramic motifs in Trypillian pottery.

Archaeology and genetics point to the modern-day territory of Romania as the origin of PCCTC. Archaeological studies link the origins of PCCTC with western Transylvania [4].

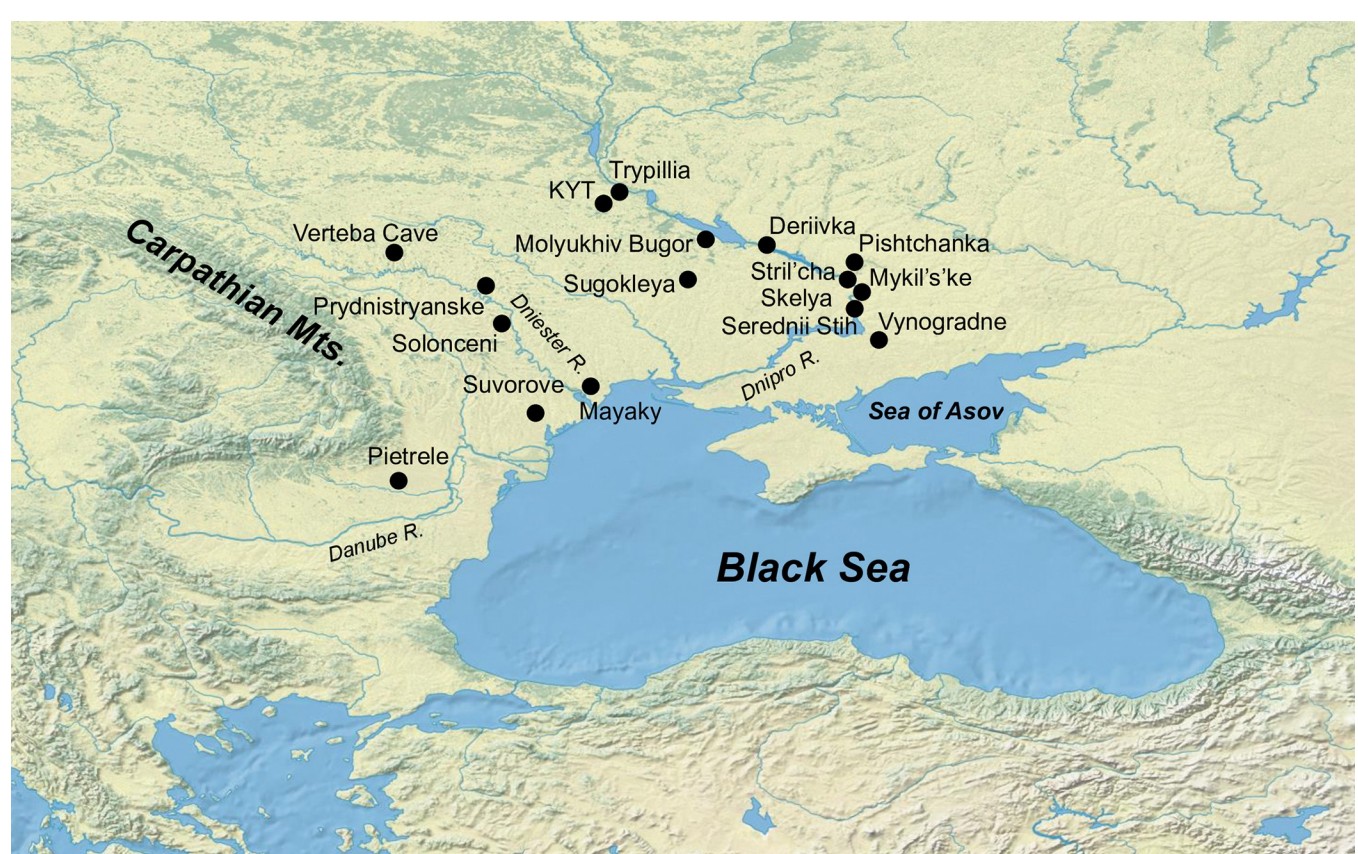

**Fig 1. Location of the Kolomiytsiv Yar Tract, as well as Trypillia, Deriivka II, Molyukhiv Bugor, Serednii Stih, Stril'cha Skelya, Pietrele, Sugokleya, Suvorove, Solonceni and Mykil's'ke archaeological sites mentioned in the text.** The background map was made with Natural Earth. Free vector and raster map data at naturalearthdata.com.

Trypillian individuals from the Verteba Cave ritual site in western Ukraine share their genetic ancestry with the Eneolithic individuals of the Bodrogkeresztúr culture from Urziceni in southeast Muntenia [5]. Trypillians maintained close genetic links with the farming communities of the Balkans through the end of the PCCTC existence [6].

Early Trypillian sites appear in the middle Dniester area at the beginning of the 5th millennium BCE, putatively reflecting eastward migration of pre-Cucuteni groups from pre-Carpathian Moldova [7]. A plausible reason behind such a migration was climate change necessitating the exploration of new territories for arable lands. This migration has been reconstructed as having taken place in several waves. Early Trypillians, having inherited the cultural and economic adaptations of the late Neolithic cultures of the Carpathian-Danube basin, managed to significantly expand the domain of ancient agricultural societies. These early east European agrarians established the basis for the formation of what Ukrainian archaeologists refer to as the Trypillian civilization between the Dniester and the Dnipro during the second half of the 5th millennium through the end of the 4th millennium BCE [8].

After 4500 BCE Trypillia spit into several local groups, distinguished by the style of their ceramics. The emergence of Cucuteni A-B—Trypillia BI-II is likely connected with the movement of PCCTC groups out of the overpopulated Carpathian region into the forest-steppe expanses of the Dniester-Dnipro interfluve. Trypillians reached the Dnipro Valley in the second half of the 5th millennium BCE. Several local Trypillian types are recognized during this period, distinguished by material culture and features of the economy. While these groups functioned autonomously, there is evidence for active interaction among them as well as with Trypillian groups to the west [9]. The transition to the open expanses of the Dnipro Valley also brought Trypillian groups into direct contact with steppe populations.

During the Trypillia BI-II stages, painted ceramics dominated in the western part of the expanding Trypillian domain, while the ceramics with grooved/incised décor dominated in the east. The eastern part of the Trypillian domain also adopted elements of pottery traditions from the steppe groups of the Serednii Stih cultural horizon in their kitchen pottery. Trypillian kitchenware is distinctive from other ceramic types by the composition of clay, which occasionally included crushed river shells, as well as by the shape of vessels (pots and bowls) and the method of applying ornamental compositions using stamping such as imprints of comb and pits (S1 and S2 Figs in S3 File). This kitchenware ceramic style is known as the "Cucuteni C" type. Cucuteni C ceramics is considered an indicator of the steppe influence on Trypillian ceramics [10].

Around 4300 BCE, the Penezhkovska-Scherbanevska local Trypillian group was formed in the middle Dnipro Valley. The group is represented by about ten settlements, of which the two largest ones are the eponymic settlement near the village of Trypillia, over 60 hectares in size, and the Kolomiytsiv Yar Tract (KYT) near the village of Kopachiv (around 15 hectares) [11] (Fig 1). Recent excavations at KYT revealed Cucuteni C ceramics in the pottery assemblage. In addition, a human long bone fragment was uncovered at the site in the Trypillian cultural layer. The presence of Cucuteni C pottery at KYT presents an opportunity to study the KYT site in the context of Serednii Stih cultural influences on Trypillia. The human osteological specimen uncovered at KYT allows the examination of diet and mobility, as well as the analysis of genetic affinities of the KYT population. In the present study, we set out to test the hypothesis of the existence of genetic interactions alongside cultural exchanges between Trypillia and Serednii Stih at Trypillian sites containing Cucuteni C pottery. Finds of human remains at Trypillian sites are exceedingly rare [12, 13]. The osteological specimen from KYT provides exceptional opportunity to expand our understanding of subsistence patterns, mobility, and genetic landscape of the area to the south of the Carpathian Mountains during the Eneolithic.

## Materials and methods

### Kolomiytsiv Yar Tract site description

Within the modern limits of the Kyiv Region, more than two dozen settlements of Trypillia stage I-II are now known. Trypillian settlements of this period, in comparison with those of the later stages BII, CI and CII, are among the least studied in central Ukraine. The settlement at the Kolomiytsiv Yar Tract (KYT) is one of the easternmost among the Trypillian sites in the northern part of this region.

Kolomiytsiv Yar Tract (50.10799, 30.52610) is situated in a valley, southeast of the village of Kopachiv, Obukhiv District, Kyiv Region (Fig 1). A stream, which is the west tributary of the Stugna River, flows southwest-northeast along the bottom of the valley (S4 Fig in S3 File). The left, sloping bank of the stream is occupied by farm fields. The steep right bank of the stream is partially overgrown with forest. The floodplain of the stream is marshy, has a width of 100 to 50–30 m, and is partially overgrown with reeds (Fig 2 and S3 Fig in S3 File).

The KYT archaeological site was discovered in 2005 [14]. It was further excavated by a joint expedition of the Kyiv Regional Archaeological Museum and the Institute of Archeology of the National Academy of Sciences of Ukraine in 2006 [15], 2007, and 2011 [16] (Field Permit number 248/0139, Authorization number 22-361/06, issued by the Institute of Archaeology, National Academy of Sciences of Ukraine, on 7/14/2006). These were rescue excavations since part of the site's cultural layer was being actively destroyed by the extraction of topsoil (chernozem) in the floodplain of a stream along which the site's Trypillian structures were found. The site is multi-layered and includes materials from the Trypillian culture (stage BI-II), the early Iron Age, and the Medieval periods (XII-XIII centuries). Archaeological monuments, later in time than the Trypillian culture, were located in the lower part of the valley, in a 70–100 m strip along the right bank of the stream.

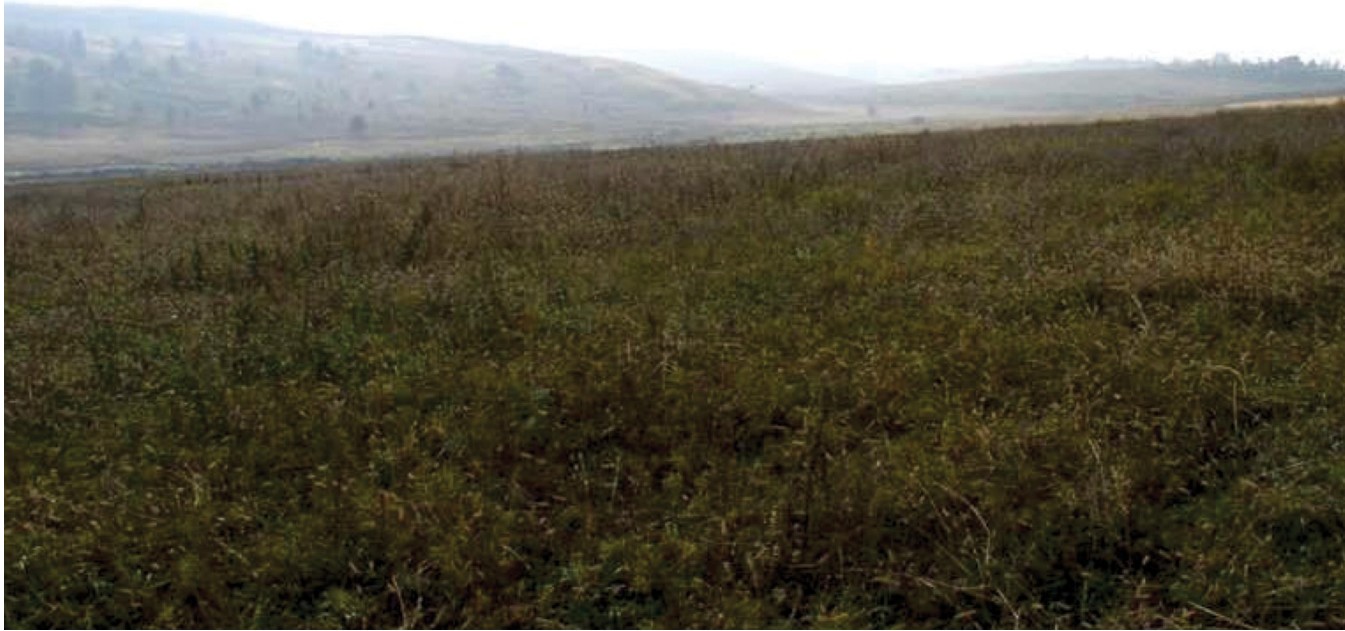

**Fig 2. The Kolomiytsiv Yar Tract setting.** Photo by M. Y. Videiko, 2006.

In cooperation with the Christian-Albrecht University, the "Human Development in Landscapes" School (Kiel, Germany), a magnetic survey was carried out on a part of the KYT's Trypillian settlement in 2016. The survey covered about three hectares of an area of the slope of the promontory in the lower part of the valley [17]. As a result, underground anomalies corresponding to the remains of 19 incinerated buildings of the Trypillian culture and traces of six pits of various sizes were found (*ibid.*, Fig 1). The survey determined that the anomalies were of varying preservation—from good to severely destroyed by plowing and erosion, especially those located in areas with a considerable slope of the terrain. Field excavations confirmed the results of the magnetic survey both regarding the presence and preservation of the underground features.

Underground features representing remains of buildings were located in three rows on the sloped terrain along the stream flowing along the bottom of the valley. The top row contained the remains of 11 buildings. The second row contained the remains of five structures, and the bottom row had three. Probably most of the remains of the buildings on this site were located outside the field, where some building remains were surveyed in 2005–2007. Pit anomalies were identified near the remains of buildings. The largest pit was located opposite the end of a building under excavation 14, which also stood out for its size. The pit's location suggests that originally these pits were used for clay extraction during dwelling construction by Trypillians.

## Analyzed specimens

In the heavily disturbed area of the topsoil extraction at the KYT archaeological site, a human long bone fragment was discovered in 2007. The bone fragment was found in the proximity of a burnt house ("*ploschadka*") of Trypillian culture, explored during excavations in 2007. The bone and buried soil specimens were subsequently sent to the Molecular Archaeology Laboratory and Archive at Grand Valley State University, following the applicable laws of Ukraine governing the export of such materials. All necessary permits were obtained for the described study, which complied with all relevant regulations.

## Radiocarbon dating and diet stable isotope analysis

Radiocarbon dating using Accelerated Mass Spectrometry (AMS) as well as carbon and nitrogen ($\delta^{13}C$ and $\delta^{15}N$) stable isotope analyses of the KYT bone fragment were carried out at BETA Analytic, Miami, FL. Stable isotope measurements were obtained using a modified version of the Longin collagen extraction method [18]. Radiocarbon calibration was carried out using OxCal version 4.4 and the IntCal 20 calibration curve [19, 20]. Stable isotope analysis of carbon and nitrogen is used in archaeological investigations to examine dietary patterns. Carbon isotope ratio value, $\delta^{13}C$, is a measure of the ratios of the stable isotopes of carbon, $^{13}C$ and $^{12}C$. Carbon isotope ratios are measured on bone collagen and dentin, which makes it possible to determine the origin of dietary proteins from marine, terrestrial and freshwater resources. Within the terrestrial sources, $\delta^{13}C$ makes it possible to differentiate between the C3 and C4 plant-based diets [21]. Nitrogen isotope ratio value, $\delta^{15}N$, is the ratios of stable isotopes of nitrogen, $^{15}N$ and $^{14}N$. The evaluation of $\delta^{15}N$ makes it possible to determine the trophic level of the studied organism within a food web. Populations of different subsistence strategies can be distinguished by their stable isotope signature [21].

Comparative analysis and graphic visualization of stable isotope data from KYT and regional population groups from the early Holocene to the Bronze Age was accomplished using Microsoft Excel 2022 for Mac, version 16.63.1.

## Strontium isotope analysis

Strontium isotope ratio analysis of the KYT bone and a buried soil sample associated with the bone was carried out at the University of Missouri Research Reactor (MURR). The soil sample was powdered in an agate mortar and calcined at 550˚C for four hours. An aliquot of about 0.1 g was transferred to a PFA vial and dissolved in 24N HF - 14N $HNO_3$ (4ml and 1ml, respectively) on a hot plate at 125˚C for 48hrs. The solution was then evaporated and the residue re-dissolved in 6ml of 6N HCl at 125˚C for 48hrs. The solution was evaporated at 90˚C, and the residue was re-dissolved in 2ml of $^{14}N$ $HNO_3$ at 125˚C and evaporated.

The bone sample was mechanically cleaned using a microdrill equipped with a bristle brush. It was then chemically cleaned using an ultra-sonic bath and a succession of 0.1N acetic acid for 30 minutes, mQ water for 15 minutes, and 0.1N acetic acid for 15 minutes. The sample was thoroughly rinsed with mQ water after each cycle. The bone was then leached for seven hours in 5% acetic acid, rinsed with mQ water, and dried at 105˚C in the drying oven. A fragment of about 40–50 mg was dissolved in a PFA vial using 7N $HNO_3$ on a hot plate at 110˚C for 24hrs. The solution was then evaporated at 90˚C.

The dry residues of the bone and the soil samples were re-digested in 2ml 7N HNO3 and the Sr was extracted using a protocol adapted from [22]. The Sr solutions were evaporated, and the residue dissolved in 0.05N $HNO_3$ before analysis.

The Sr isotopic analysis was conducted on a Nu Plasma II (Nu Instruments) multi-collector–inductively coupled plasma–mass spectrometer in operation at MURR. Both the samples and the SRM987 Sr isotopic standard solutions were prepared to obtain a Sr concentration of about 150 ppb. The measured ratios were corrected for the isobaric interference of $^{87}Rb$, $^{86}Kr$, and $^{84}Kr$, and for mass bias using the iterative approach and a value of 0.1194 for $^{86}Sr/^{88}Sr$ natural ratio. The SRM987 was measured multiple times (n = 11) and the value obtained for the $^{87}Sr/^{86}Sr$ were 0.71022 ± 0.00004 (2sd). The values obtained for the samples were corrected by standard bracketing using the accepted value of 0.710248 for the $^{87}Sr/^{86}Sr$ [23].

## Genetic analysis

We carried out the first steps of ancient DNA analysis in a dedicated clean room at Harvard Medical School, following previously established protocols to minimize contamination including working in positively pressured rooms decontaminated with ultraviolet light, and use of protective clothing by the technicians handling the remains. We took a sample of cortical bone powder using a dental drill. We extracted DNA using a methodology optimized to retain short and degraded molecules [24–26], and converted the extracted DNA into a barcoded, double-stranded library [27]. We enriched the library in solution for sequences overlapping at least 1.2 million single nucleotide polymorphisms [28, 29]. We sequenced on Illumina instruments using 2x76bp sequences. We processed the data bioinformatically and mapped to the human reference genome hg19 as described in previous studies, removed duplicated molecules based on matching barcodes and start and stop position in the mapped genome (e.g., [30]). We represented each targeted position by a single randomly chosen sequence, which produced a total of 76612 targeted SNPs on chromosomes covered by at least one sequence. The sequence data gave no evidence of contamination based on polymorphism in its mitochondrial DNA using contamMix [31] (the 95% confidence interval for matching to the consensus haplogroup of U4b1b2 was 98.9–100%) and had ratio of Y to X chromosome sequences consistent with female molecular sex.

## Inclusivity in global research

Additional information regarding the ethical, cultural, and scientific considerations specific to inclusivity in global research is included in the S1 File.

## Results and discussion

### Ceramic assemblage at Kolomiytsiv Yar Tract

Detailed analysis of the KYT ceramic assemblage is presented in [17]. Overall, Trypillian ceramics at the KYT site belongs to the stage I-II of Trypillia culture period. The ceramic assemblage uncovered at the site is typical for this stage in the Middle Dnipro region and is represented by the so-called "kitchenware" and "tableware" types. A characteristic feature of the ceramics from KYT is the addition of crushed shells to the clay of some vessels with a carved ornament, as well as the processing of their inner surface with a tool that leaves stripes —combs, which is typical for ceramics of the Cucuteni C type (S1 and S2 Figs in S3 File). A detailed comparative analysis of ceramics of the Cucuteni C type from the settlements of Trypillia and Serednii Stih convincingly shows that Trypillian settlements contain imported from Serednii Stih as well as locally made ceramics with shell admixture [32]. Finds of Cucuteni C ceramics at KYT are of local production. Cucuteni C pottery at KYT parallels the Serednii Stih ceramics of Serednii Stih II, Stril'cha Skelya and Molyukhiv Bugor (S2 Fig in S3 File). The ceramic assemblage at KYT containing Cucuteni C ceramics made on site suggests the presence of carriers the Serednii Stih pottery making techniques at KYT.

The presence of Cucuteni C ceramic style at KYT suggests the interaction of the Trypillian population of KYT with and influence by Serednii Stih. The two most proximal to KYT Serednii Stih settlements on the right bank of the Dnipro River are Deriivka II and Molyukhiv Bugor (Fig 1). The Serednii Stih layer at the Molyukhiv Bugor site is dated to 3951–3640 BCE [33]. Radiocarbon dates from the Deriivka II settlement place it in 4436–3988 BCE [34]. It has been suggested that the formation of culture groups of the Serednii Stih II period (according to D. Telegin's classification, [35]), represented by settlements at Molyukhiv Bugor and Deriivka II, was influenced by Trypillia [1, 2, 36]. Archaeological evidence suggests that the Serednii Stih populations at Deriivka II and Molyukhiv Bugor practiced agriculture, but their main form of subsistence was hunting and pastoralism, supplemented by freshwater fish and mollusks [1, 2, 34, 37, 38]. The presence of Serednii Stih groups in the forest steppe zone indicates the movement of Serednii Stih northward during the Serednii Stih II period.

### Absolute dating and dietary isotopes of the KYT individual

Kolomiytsiv Yar Tract human osteological specimen was directly dated to 5170±30 BP (Table 1). After calibration, this corresponds to 4049–3820 calBCE (95.4% probability). Thus, radiocarbon analysis confirms the placement of the specimen within the Trypillian BI-BII period.

The $\delta^{15}N$ of 12.0‰ suggests the potential presence of Freshwater Reservoir Effect (FRE) from aquatic dietary sources [33, 34]. The $\delta^{15}N$ terrestrial baseline is between 9 and 11‰. Past 12‰, the FRE can produce a radiocarbon reservoir offset of 250+ years [39, 40]. Considering the $\delta^{15}N$ of the KYT specimen being just over the terrestrial baseline threshold, the FRE influence on the date is likely at the lower end of the FRE influence scale. It is not possible to quantify the FRE more precisely in the absence of contemporaneous faunal remains.

**Table 1. Radiocarbon and stable isotope data of the Kolomiytsiv Yar Tract samples.**

| Sample | Date, uncalBP (lab code) | Date, calBCE[a], 95.4% probability | $\delta^{13}C$ (‰) | %C | $\delta^{15}N$ (‰) | %N | C:N ratio | 87Sr/86Sr | 87Sr/86Sr 2se |
|---|---|---|---|---|---|---|---|---|---|
| Human bone | 5170±30 (Beta– 523816) | 4049–3945 (94.7%); 3831–3820 (0.8%) | -20.1 | 42.1 | 12 | 15.01 | 3.1 | 0.71109 | 0.00001 |
| Buried soil | | | | | | | | 0.73320 | 0.00001 |

[a] Calibration was performed using IntCal20 calibration curve [19, 20].

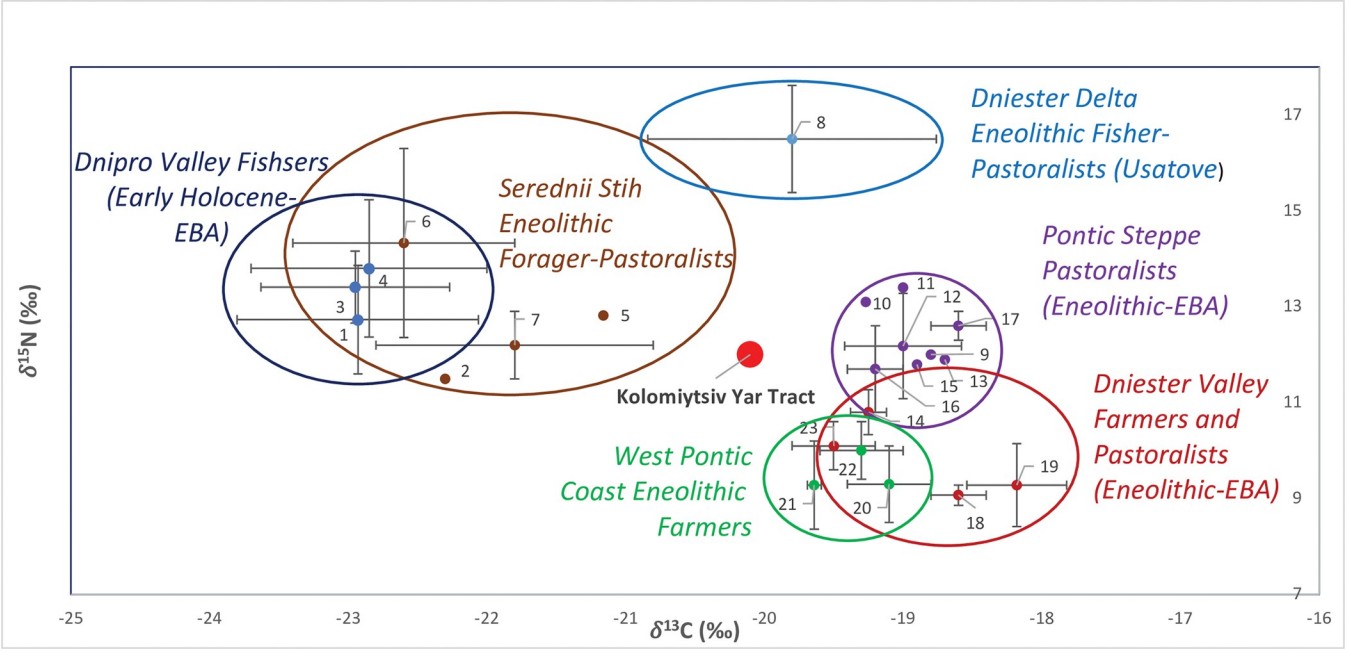

**Fig 3. Distribution of δ¹³C and δ¹⁵N stable isotope ratios of the early Holocene-Early Bronze Age (EBA) populations of the North and West Pontic area.**
Stable isotope ratios and corresponding publication sources are listed in S1 Table in S2 File. 1, Deriivka I, III; 2, Deriivka II; 3, Mykil's'ke; 4, Yasynuvatka; 5, Oleksandria; 6, Igren 8; 7, Molyukhiv Bugor; 8, 12, Mayaky; 9, 13, Pishtchanka; 10, Kam″yanka-Dniprovs'ka; 11, 17, Vynogradne; 14, Pidlisivka/Porohy; 15, Shakhta Stepna; 16, Sugokleya; 18, 19, Prydnistryanske; 20, Durankukak; 21, Smyadovo; 22, Varna; 23, Verteba Cave.

Stable isotope values of the KYT specimen were plotted against the corresponding values from Ukrainian Neolithic, Eneolithic and Early Bronze Age populations for which stable isotope data are currently available (Fig 3 and S1 Table in S2 File). Kolomiytsiv Yar Tract specimen's values for both δ¹³C and δ¹⁵N were outside of the range of variation of the Trypillian farming groups from Verteba Cave [41] and Prydnistryanske of the Yampil complex [42] (Fig 1).

Nitrogen δ¹⁵N ratio of KYT was within the range of variation of the Eneolithic-EBA Pontic steppe pastoralists, Eneolithic forager pastoralists from the Serednii Stih horizon, as well as the Dnipro Valley fishers, at the lower range of δ¹⁵N ratio variation for the latter group (Fig 3). Carbon δ¹³C ratio of the KYT specimen was within the range of variation of the Dniester Delta Eneolithic population from the Usatove site at Mayaky, with subsistence based on fishing and pastoralism [40, 43], and outside the range of δ¹³C variation for steppe pastoralists. Based on diet isotope values, we conclude that KYT may potentially originate from a population with the subsistence strategy based on foraging and/or pastoralism.

At the same time, we cannot exclude that the particular isotope ratios found in the KYT individual could be a result of some special diet. The diet of the KYT individual could also be reflecting a mixture of farming, pastoralist and foraging subsistence practices, if KYT were a forager living in a farming environment or vice versa.

## Strontium isotope analysis

The ⁸⁷Sr/⁸⁶Sr ratio value measured in the KYT bone specimen was 0.71109 and the one measured in the buried soil was 0.73320 (Table 1). No samples were collected to define a strontium isotopic baseline for the purpose of the present study. The strontium isotopic composition of the bone and soil samples were compared to data from [44] for estimated values of the ⁸⁷Sr/⁸⁶Sr ratio values across the different geological ages in Ukraine. These estimations were

derived from values presented in [45]. The geological setting of the KYT site and the adjacent stream is mainly composed of sedimentary rocks and sediments from the Eocene, Oligocene and Miocene, with estimated range of $^{87}$Sr/$^{86}$Sr ratio values between 0.709 and 0.711 [44]. Rocks of Precambrian age with estimated range of values $^{87}$Sr/$^{86}$Sr ratio values between 0.712 and 0.780 [44] can be found West and South within a range of 35–45 km from the KYT site. The lithologies around the KYT site are presented in detail in S4 Fig in S3 File.

The $^{87}$Sr/$^{86}$Sr ratio value of the soil sample (0.73320) is higher than the range proposed for the Cenozoic substrate present at the KYT site. No strontium isotopic data on the specific lithologies developed at the site is available, which makes it difficult to assess whether the soil is representative of the substrate or if strontium from another source has contaminated the soil. The site itself, and the area drained by the adjacent stream, are not currently farmed, but agriculture is common in the region and the application of fertilizers is likely. Numerous studies have demonstrated the impact of modern fertilizers on the strontium isotopic signature of soils (e.g., [46–48]) and archaeological remains buried in these soils (e.g., [49]). Available data for the strontium isotopic composition of fertilizers exhibit lower values than that of the soil sample from KYT (e.g., [47, 48, 50]). However, some data for K-fertilizers also demonstrates that they can present much higher values (e.g., [47, 51]). Additional data, including the identification of the types of fertilizers used in the region, would be required to determine if the strontium isotopic signature from the KYT soil has been affected by the use of K-fertilizers. No older rock of Proterozoic or Archean age with higher strontium ratio value is present along the stream adjacent to the site (S4 Fig in S3 File), and all lithologies around the site are of Cenozoic age and exhibit lower strontium ratio values than the soil sample. Further sampling around the site would be needed to define the strontium isotopic signature of the site and its immediate surroundings.

The KYT bone sample $^{87}$Sr/$^{86}$Sr ratio value is slightly higher than the range of estimated values for the Cenozoic substrates. The signature in the bone could be the result of a mixture of Cenozoic substrates with an input from Precambrian rocks that present higher strontium ratio values. This would be consistent with the diet of a forager-pastoralist who would get their food from the area of Precambrian rocks to the South and West from the KYT site. However, these substrates, and regions where they can be found in association, have a broad distribution, which prevents precise association of the KYT bone specimen with a specific area or site.

The KYT bone sample's strontium ratio was also compared to the signature obtained from human enamel from individuals of the Eneolithic and Yamna populations of Ukraine [44, 52]. The KYT individual's strontium ratio falls within the range of values observed for the site of Sugokleya (Kirovograd Region), which is associated with steppe cultures of the Eneolithic and the Bronze Age [52]. More specifically, the KYT bone's strontium ratio is close in value to that from the enamel of an adult individual from the Sugokleya Kurgan Grave 10 (UK 44/45, [52], S5 Fig in S3 File). The Sugokleya site is located in the vicinity of the Molyukhiv Bugor and Deriivka II settlements of the Serednii Stih culture in the Middle Dnipro Valley (Fig 1). At the same time, the present data do not allow to unequivocally identify the place of origin of the KYT individual nor to firmly establish whether the individual was non-local to the KYT site.

The fact that bone is a porous tissue and its biogenic strontium signature is commonly affected by diagenesis/chemical exchanges within the burial environment has consequences for this research. Because the soil in immediate contact with the bone is higher than that of the bone, it is possible that the value of the bone was originally lower than the value that was measured.

Taken together, diet isotope analysis of the KYT individual suggests the forager-pastoralist subsistence type such as that of forest-steppe Serednii Stih communities of the middle Dnipro Valley. Strontium isotopic signature of the KYT individual is compatible with a large

geographic area, encompassing most of the Middle Dnipro Valley, and close to the strontium isotope ratios in the vicinity of the Molyukhiv Bugor and Deriivka II settlements of Serednii Stih (Fig 1). The presence of Cucuteni C pottery at the KYT site pointing at the presence of Serednii Stih pottery makers at KYT [11] and the similarity of ceramic assemblages between KYT and Serednii Stih, further support the origin of the KYT individual from a Serednii Stih population.

### Genetic analysis of the KYT specimen

To examine the genetic ancestry of the KYT individual, we extracted and examined the KYT individual's DNA. Genetic analysis revealed that the individual was genetically a female, carrying mtDNA haplogroup U4b1b2 (Table 2). Carriers of the U4b subclade were identified in Mesolithic and Neolithic fishers and foragers from Ukraine, the Iron Gates area of the Danube, and the Baltic coast [5]. The U4 mtDNA clade was present in the Middle Dnipro Valley from the beginning of the Holocene and persisted in the North Pontic steppe through at least the late Eneolithic [5, 53–55]. Carriers of the U4b mtDNA lineage have not been identified in Trypillian remains studied to date. At the same time, a late Trypillian individual dated to 3482–3297 calBCE from the Gordineşti site in Moldova and identified as having a substantial steppe genetic admixture, carried a U4-derived mitochondrial lineage U4a1 [6]. On the genome-wide Principal Component Analysis (PCA), the Gordineşti individual is pulled away from the cluster of Trypillian specimens from Verteba cave, dated to ca. 3900–3600 calBCE, towards the EBA Yamna individuals from Ukraine and the Volga River region.

Enrichment of a DNA derived from the KYT specimen for a set of more than 1.2 million single nucleotide polymorphisms (SNPs) across the genome ('1240K enrichment') yielded data from 76612 Single Nucleotide Polymorphisms, enough to permit high-resolution analysis of the ancestral affiliations of this individual (Table 2 and S2 Table in S2 File). We carried out Principal Component Analysis (PCA) on modern West Eurasians of approximately 600,000 autosomal SNPs genotyped on the Affymetrix Human Origins array (Fig 4). These are a subset of those targeted by 1240k enrichment, and we had data for 41,308 of these SNPs for the KYT individual, enough to permit high-resolution ancestral analysis. Many ancient samples fall outside the genetic range of modern Eurasia. The KYT sample (marked as SSX in Fig 4) falls close to Yamna pastoralists of the EBA. KYT-SSX has more ancestry from the ancient steppe than any modern West Eurasian sample, but is very distinct from Mesolithic-Neolithic hunter-gatherers, either Eastern Hunter Gatherers (EHG) or even more so samples from the Mesolithic in Serbia (Iron Gates) which are genetically primarily of Western Hunter Gatherer (WHG)-associated ancestry.

While the KYT-SSX specimen clusters most closely with Yamna, both from Ukraine (Shevchenko) and Russia (Middle Volga, Samara Region), it does not form a clade with the Yamna. An $f_4$ test [56] for $f_4$ *(Ancient African genomes, Serbia Mesolithic; SSX, Ukraine EBA Yamnaya)* produced a $Z$ score of −3.85, and for Samara Yamna instead of KYT-SSX $Z$ = −4.78. Statistical analysis of whole-genome data further revealed that KYT-SSX was not genetically similar to Trypillia individuals studied to date. Thus, for example, for $f_4$ *(Old Africa, Turkey Neolithic; SSX, Trypillia)* the $Z$-score is 6.692. A plausible scenario is that the KYT individual has much of their ancestry from a Proto-Yamna population [57], while also being admixed with Serbia/

**Table 2. Genomic data of the Kolomiytsiv Yar Tract sample.** Full DNA library report is presented in S2 Table in S2 File.

| Specimen/DNA lab code | Molecular sex | MtDNA haplotype | SNP hits on autosomal targets | Coverage on autosomal targets |
|---|---|---|---|---|
| Human long bone /I7585 | Female | U4b1b2 | 76612 | 0.068193 |

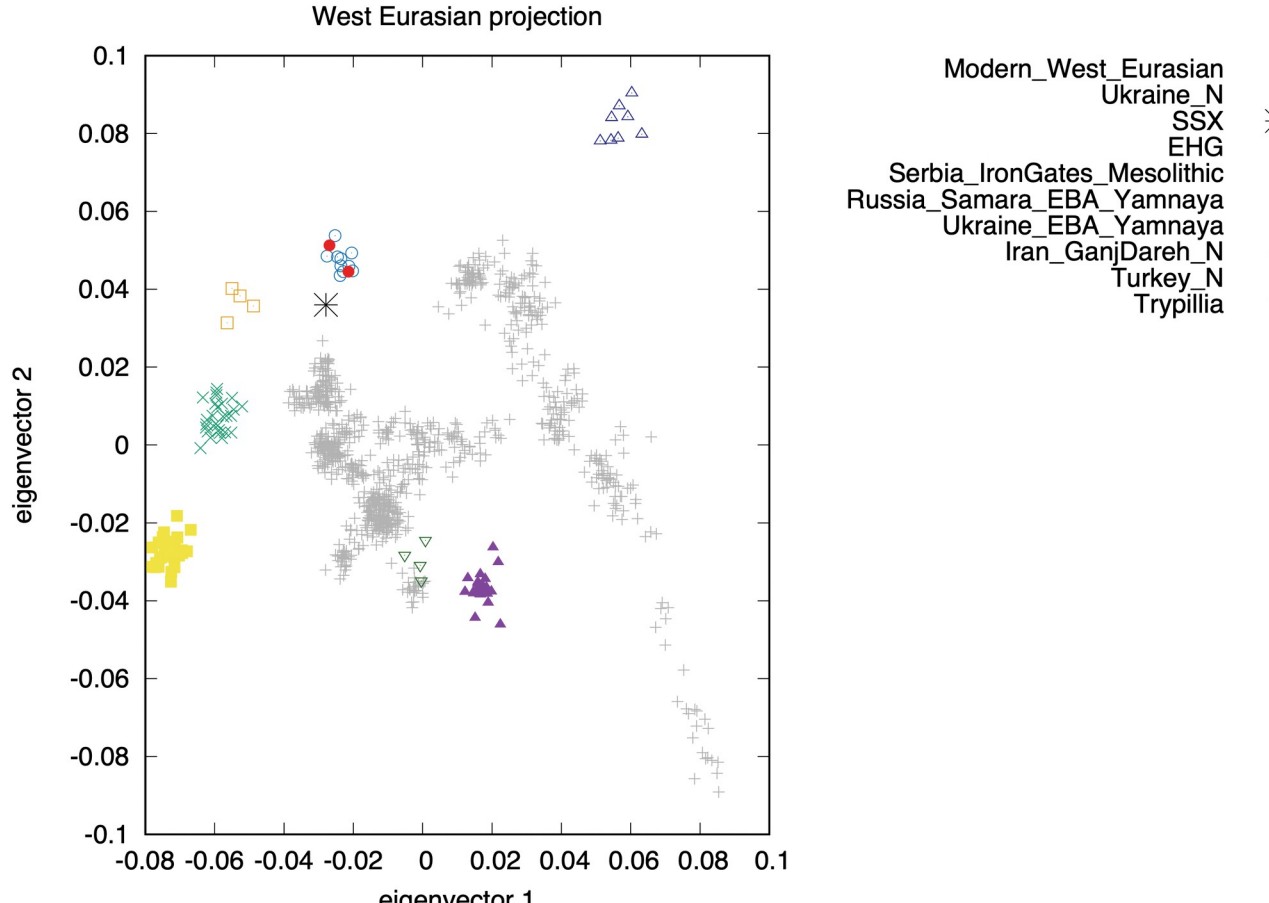

**Fig 4. PCA projection of whole genome data of published West Eurasian prehistoric and modern (grey points) populations.** Ukraine_N, Ukrainian Neolithic; SSX, KYT specimen; EHG, Eastern Hunter Gatherers; Iran_GanjDareh_N, Iranian Neolithic; Turkey_N, Anatolian Neolithic.

Iron Gates Mesolithic, the latter possibly coming from the Neolithic populations of the Dnipro Valley that have been shown to carry WHG admixture [5]. Serednii Stih is the main Proto-Yamna group of the Eneolithic North Pontic steppe and has been hypothesized to be ancestral to Yamna based on archaeological analysis [2]. The similarity in genetic ancestry between a Serednii Stih individual from the Deriivka II site and Yamna individuals from Samara was previously reported [53]. Taken together, the archaeological and genetic evidence discussed above is consistent with the assignment of the KYT individual to deriving most of its ancestry from Serednii Stih people.

The presence of an individual with Yamna-related ancestry, most plausibly derived from Serednii Stih groups, at the Trypillian KYT site proposes a possibility of genetic interactions between the steppe and farming communities as early as the late 5$^{th}$–early 4$^{th}$ millennium BCE. However, evidence for such interactions is currently lacking. As mentioned above, Trypillians studied to date did not carry steppe genetic admixture until after 3482–3297 calBCE. At the same time, details about Trypillian ancestry of the first part of the 4$^{th}$ millennium come from a single site (Verteba Cave) and a limited number of specimens. Nevertheless, the widespread presence of Cucuteni C pottery at Trypillian sites of the early-middle period, including Verteba, suggests the Serednii Stih-Trypillia interactions were extensive, thus providing multiple opportunities for biological interactions between the two culture groups as well. Further

examination of genomes from Serednii Stih and Trypillia sites would help clarify the timing of genetic interactions between Trypillians and Eneolithic steppe forager-pastoralists.

## Conclusions

The archaeological, stable isotope and genetic analyses presented in this report produce the following life history highlights for the KYT individual. While the bone fragment of the KYT individual was found in Trypillia culture context, the individual was not genetically associated with the ancestral pool of European Neolithic farmers from which Trypillian ancestry is derived. The KYT individual, genetically a female, is a representative of a Proto-Yamna population of Eneolithic forager-pastoralists of the North Pontic area, such as Serednii Stih. The individual's maternal (mtDNA) lineage stems from an autochthonous to the North Pontic area clade, which was present in the Dnipro Valley since the beginning of the Holocene. The individual's diet was consistent with that of a forager-pastoralist, or the individual had a mixed diet, potentially including resources coming from foraging, pastoralism and farming. The origin of the KYT individual is likely in the middle Dnipro Valley, plausibly from a location in the vicinity of the Molyukhiv Bugor and Deriivka II settlements of Serednii Stih.

Integrating the data presented in this report with the existing body of archaeological knowledge presents the following picture of the population dynamics at the end of the 5th millennium in the East Balkans—North Pontic area. Archaeological evidence indicates the existence of contacts between Serednii Stih, PCCTC and KGK communities of the West Pontic from before 4700–4600 BCE through ca. 4200 BCE, during the operation of the Eneolithic Circum-Pontic trade network [58]. The spread of ceramic finds and copper products allows us to reconstruct the trade route from the western part of KGK culture complex in the Lower Danube Valley (the Pietrele site in Romania) to the Bugeac steppe in the Northwest Pontic (e.g. the site of Suvorove), following north-east to the Middle Dniester PCCTC groups of the Solonceni-Zalischyky type, and then further east to the Dnipro River (Fig 1). Imports of Trypillia BI ceramics at the settlement of Serednii Stih and other archaeological sites in the Lower Dnipro Valley including Mariupol-type cemeteries such as Mykil's'ke (Nikolskoye) (Fig 1) testify to the extent of these trade connections to the Dnipro Rapids area. This trade route operated for more than 500 years [59]. After 4200 BCE, following the collapse of the Balkan Eneolithic communities in the result of climatic changes [58] that also affected the steppe belt north of the Black Sea, the Serednii Stih populations moved from the steppe to forest steppe along the Middle Dnipro Valley, where they came into direct contact with PCCTC. This contact is evidenced by the spread of Cucuteni C ceramics, patterned after the steppe ceramics, in Trypillian settlements. It has been suggested that the Cucuteni C ceramic style manifests the presence of the bearers of steppe ceramic technology at Trypillian settlements [11]. The population affiliation of the KYT individual with Proto-Yamna/ Serednii Stih and the presence at KYT of Cucuteni C ceramics made on site not only corroborates the presence of Serednii Stih at Trypillian sites, but also reflects the integration of steppe individuals into Trypillian societies. This study exemplifies a case when the hypotheses of archaeologists, based on the use of traditional typological and comparative methods, received strong validation from molecular analysis methods employed by natural sciences.

## Supporting information

**S1 File. Inclusivity in global research statement.**
(DOCX)

**S2 File.**

(XLSX)

**S3 File.**

(PDF)

## Acknowledgments

The authors would like to thank Maxim Kvitnicky for his archaeological expertise during the excavations at KYT, as well as Nadin Rohland, Swapan Mallick, Matthew Mah, Adam Micco, and Aisling Kearns for their skillful preparation and processing of the KYT sample for ancient DNA analysis. This article is subject to the HHMI's Open Access to Publications policy. HHMI lab heads have previously granted a nonexclusive CC BY 4.0 license to the public and a sublicensable license to HHMI in their research articles. Pursuant to those licenses, the author-accepted manuscript of this article can be made freely available under a CC BY 4.0 license immediately upon publication.

## Author Contributions

**Conceptualization:** Alexey G. Nikitin.

**Data curation:** Mykhailo Videiko, Nick Patterson, David Reich.

**Formal analysis:** Alexey G. Nikitin, Nick Patterson, Virginie Renson.

**Funding acquisition:** Alexey G. Nikitin, David Reich.

**Investigation:** Mykhailo Videiko.

**Methodology:** Alexey G. Nikitin, Nick Patterson, Virginie Renson.

**Project administration:** Alexey G. Nikitin.

**Resources:** David Reich.

**Writing – original draft:** Alexey G. Nikitin, Nick Patterson, Virginie Renson.

**Writing – review & editing:** Alexey G. Nikitin, Mykhailo Videiko, Virginie Renson, David Reich.

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
