## [Decision Letter · Decision Letter 0]

14 Mar 2023

PONE-D-22-34359Interactions between Trypillian farmers and North Pontic forager-pastoralists in Eneolithic central UkrainePLOS ONE

Dear Dr. Nikitin,

Thank you for submitting your manuscript to PLOS ONE. After careful consideration, we feel that it has merit but does not fully meet PLOS ONE’s publication criteria as it currently stands. Therefore, we invite you to submit a revised version of the manuscript that addresses the points raised during the review process. Dear Authors, 

your manuscript was evaluated by two reviewers. They both found it very interesting and did not identify any major problems with the ms. Reviewer 2 provided some suggestions which may improve the quality of the ms. Please consider these suggestions but they are not obligatory. He asked also to clarify the discrepancy in reported numbers of genomic SNPs and suggested that authors should be more cautious with the interpretation of isotopic results. I deem it necessary to address these two small issues before publication.

Best regards, Please ensure that your decision is justified on PLOS ONE’s publication criteria and not, for example, on novelty or perceived impact.

We look forward to receiving your revised manuscript.

Kind regards,

Mateusz Baca

Academic Editor

PLOS ONE

Journal Requirements:

2. Please include a complete copy of PLOS’ questionnaire on inclusivity in global research in your revised manuscript. Our policy for research in this area aims to improve transparency in the reporting of research performed outside of researchers’ own country or community. The policy applies to researchers who have travelled to a different country to conduct research, research with Indigenous populations or their lands, and research on cultural artefacts. The questionnaire can also be requested at the journal’s discretion for any other submissions, even if these conditions are not met.  Please find more information on the policy and a link to download a blank copy of the questionnaire here: https://journals.plos.org/plosone/s/best-practices-in-research-reporting. Please upload a completed version of your questionnaire as Supporting Information when you resubmit your manuscript

3. In your manuscript, please provide additional information regarding the specimens used in your study. Ensure that you have reported specimen numbers and complete repository information, including museum name and geographic location.

For more information on PLOS ONE's requirements for paleontology and archeology research, see https://journals.plos.org/plosone/s/submission-guidelines#loc-paleontology-and-archaeology-research.

Natural Earth (public domain): http://www.naturalearthdata.com

Reviewers' comments:

Reviewer's Responses to Questions

**Comments to the Author**

1. Is the manuscript technically sound, and do the data support the conclusions?

Reviewer #1: Yes

Reviewer #2: Yes

2. Has the statistical analysis been performed appropriately and rigorously? 

Reviewer #1: Yes

Reviewer #2: Yes

3. Have the authors made all data underlying the findings in their manuscript fully available?

Reviewer #1: Yes

Reviewer #2: Yes

4. Is the manuscript presented in an intelligible fashion and written in standard English?

Reviewer #1: Yes

Reviewer #2: Yes

5. Review Comments to the Author

Reviewer #1: The relation between Tripolje complex farming communities and advanced foragers belonging to Stredny Stog has been contested. The archaeology speaks of cultural exchange in pottery as well as certain farming practices. Since we lack human bones, a single bone is here undergoing a full interdisciplinary treatment to illuminate its origin, which points to Stregny Stog thus confirming archaeological evidence of contacts.

The paper provides a full contextual background to the archaeological problem, and provides as well a full methodological presentation of all analyses. AN exemplary presentation.

The conclusion could have been more bold, providing some tentative hypotheses as to the nature of this interaction, whether peaceful e.g. trade or rather captives. Or something in between.

Reviewer #2: Nikitin and colleagues present a joint analysis of isotopic and genetic data obtained from one individual from the Kolomiytsiv Yar Tract (KYT) in central Ukraine, which belongs to the Trypillian culture (the eastern part of the Cucuteni-Trypillia complex). During the late Neolithic, the Cucuteni-Trypillia complex occupied a large geographic area, encompassing major parts of present-day eastern Romania, Moldova, and Ukraine, and is characterised by the emergence of settlement mega-sites with several hundreds of inhabitants. Yet, skeletal material from the Trypillian culture is rare and thus only limited DNA data was so far produced. To my knowledge, genome-wide information of only 20 individuals from Ukraine, and four individuals from Moldova are currently available. Consequently, this study provides a relevant addition to the current archaeogenetic record. According to the novel radiocarbon date reported in this study, the analysed individual belongs to the Trypillia BI-II stage, which is associated in the east with the introduction of pottery traditions potentially adopted from neighbouring steppe pastoralist groups, especially the so-called “Cucuteni C ceramics”. Since this kind of pottery is also found at the KYT, indicating some cultural exchange between the local Trypillian population and the adjacent steppe pastoralists, the authors investigate whether this cultural signal is also reflected in genome (and isotopic signature) of the reported individual. For this purpose, the authors report novel carbon and nitrogen data to reconstruct the diet of the individual as well as strontium and genome-wide DNA data to trace the origin and ancestry of the specimen. Indeed, the nitrogen and carbon isotope values indicate that the individual’s diet was different to the diet of the local Eneolithic farmers but is shifted in the direction of the fisher-forager-pastoralists of the North Pontic steppe. The authors highlight that the diet of the individual could also reflect “a mixture of farming, pastoralist and foraging subsistence practice”. Genetically, the individual is remarkable since it clusters outside the genetic variation of previously published Trypillia samples, which carry mainly Early European Farmer ancestry, and clusters closely with early Bronze Age Yamnaya individuals, proving gene low from the steppe pastoralists into Trypillia farmer communities. Consistently, the strontium isotope values of the bone sepcimens fall in the range of previously published Yamnaya individuals. Yet, the individual is genetically not cladal with Yamanya individuals (as shown using F4 statistics), suggesting that the individual is ultimately admixed between a major Steppe source and a minor WHG enriched source.

In my opinion, the paper combines comprehensibly different lines of evidence to present a convincing story. The authors provide the first individual with Trypillia cultural affiliation that carries a majority of Steppe-derived ancestry, which is a great addition to the archaeogenetic record (especially for a geographic region in a period that is so far understudied) and service to the aDNA community. As far as I can assess the methodology, the analyses were conducted according to the scientific standard. The laboratory protocols for DNA and isotope extraction and processing are described in detail and cite the relevant references. Before population genetic analysis, data quality was ensured by measuring contamination on the mtDNA level. The authors applied well-established population genetic methods to the genome-wide data, namely PCA and F4-statistics. For both the isotopic measurements and population genetic methods, all relevant results were reported (Standard deviations for isotope values, Z-values for F4-statistics, etc). Therefore, I consider this paper to be technically sound.

My comments regarding the scientific quality are as follows:

The paragraph “Ceramic assemblage at Kolomiytsiv Yar Tract” (lines 257 to 270) in the Results and Discussion section does not report new results, so maybe it would be more appropriate to move its content to the introduction.

In the lines 306 to 307, the authors write “Overall, the KYT diet appears comparable to the diet of fisher-forager-pastoralists of the North Pontic steppe, but not to the diet of farmers of Eneolithic Pontic coast and adjacent forest steppe areas (Figure 3)”. In my opinion, this is a contestable statement as the sample plots between all six reference groups and does not cluster with any of them specifically. While the authors also state that “the diet of the KYT individual could also be reflecting a mixture of farming, pastoralist and foraging subsistence practices”, which mitigates the previous assertion, I would advocate for a more cautious conclusion.

In the segment “Strontium isotope analysis” in the Results and Discussion section (lines 331 to 397), the authors reference several previously published 87Sr/86Sr ratios for Eneolithic and Yamnaya populations from Ukraine. Without knowledge of these publications, it was a difficult for me to follow, thus I wonder if it would be possible to summarise the strontium values in a similar form as it was done for the nitrogen and carbon values in Figure 3.

In line 416 the authors write “Whole genome analysis of the KYT specimen produced 76550 Singe Nucleotide Polymorphism “hits” (…)”. I would suggest to rephrase that since it is not clear to me what “whole genome analysis” means in this context. I guess this describes “sequencing of the aDNA libraries after 1240k target enrichment”. In general, since the data is SNP captured and not shotgun sequenced, I would advocate for the use of the term “genome-wide data” instead of “whole genome data”. Also, I cannot not reconstruct why only 76550 SNPs were genotyped, when (according to lines 247 to 248) “a total of 357,145 targeted SNPs on chromosomes [were] covered by at least one sequence”. Additionally, I would like to know the number of SNPs available for PCA. In Table S2, only the 1240k SNPs are reported, yet the PCA was run on Human Origins data, which is only a subset of the 1240k SNP set.

In lines 432 to 436, the authors write “A plausible scenario is that the KYT individual has much of their ancestry from a Proto-Yamna population (Chintalapati et al. 2022), while also being admixed with Serbia/Iron Gates Mesolithic, the latter possibly coming from the Neolithic populations of the Dnipro Valley that have been shown to carry WHG admixture (Mathieson et al. 2018)”. The paper could highly profit from formally testing this hypothesis, e.g. using qpAdm. While PCA and F4-statistics clearly suggest a major proportion of Steppe ancestry in the KYT-SSX individual, qpAdm testing would provide a (citable) percentage and could be furthermore applied to identify the source population that provided the smaller ancestry proportion (which causes the shift away from Yamnaya in the PCA).

I have also noticed some smaller typos which should be fixed after another round of proof-reading, e.g.:

Line 56 „The Yamma cultural (…)“

Line 237 „We took a sample of xx”

In Figure 3 “Dnipro Valley Fihsers” and “Dniester Delta Eneolithic Fiher”

Summarising, I would like to see this study published and think that after some small adjustments the manuscript should be ready for publication.

6. PLOS authors have the option to publish the peer review history of their article (what does this mean?). If published, this will include your full peer review and any attached files.

Reviewer #1: **Yes: **kristian kristiansen

Reviewer #2: No

---

## [Author Response · Author response to Decision Letter 0]

3 Apr 2023

April 2, 2023

Re: PLOS ONE Decision: Revision required [PONE-D-22-34359] 

Dear Dr. Baca,

Thank you for the opportunity to revise our manuscript. We appreciate the thoughtful comments provided by the reviewers.

We modified the manuscript to address the following two issues raised in your editorial comments and also as the response to Reviewer #2:

1) We clarified the discrepancy in reported numbers of genomic SNPs. 

2) We re-wrote the interpretation of isotopic results (please see below for details). 

In addition, we modified the manuscript to incorporate the suggestions raised by Reviewer #2 as elaborated below.

Reviewer 2:

“In the lines 306 to 307, the authors write “Overall, the KYT diet appears comparable to the diet of fisher-forager-pastoralists of the North Pontic steppe, but not to the diet of farmers of Eneolithic Pontic coast and adjacent forest steppe areas (Figure 3)”. In my opinion, this is a contestable statement as the sample plots between all six reference groups and does not cluster with any of them specifically. While the authors also state that “the diet of the KYT individual could also be reflecting a mixture of farming, pastoralist and foraging subsistence practices”, which mitigates the previous assertion, I would advocate for a more cautious conclusion.”

Response: We re-wrote the paragraph to produce a more cautious assessment of the subsistence practices of the KYT individual and removed the sentence mentioned by the Reviewer. 

“In the segment “Strontium isotope analysis” in the Results and Discussion section (lines 331 to 397), the authors reference several previously published 87Sr/86Sr ratios for Eneolithic and Yamnaya populations from Ukraine. Without knowledge of these publications, it was a difficult for me to follow, thus I wonder if it would be possible to summarise the strontium values in a similar form as it was done for the nitrogen and carbon values in Figure 3.”

Response: We have added S5 Fig containing a data plot of the strontium ratios of the Eneolithic and Yamna populations from Ukraine.

“In line 416 the authors write “Whole genome analysis of the KYT specimen produced 76550 Singe Nucleotide Polymorphism “hits” (...)”. I would suggest to rephrase that since it is not clear to me what “whole genome analysis” means in this context. I guess this describes “sequencing of the aDNA libraries after 1240k target enrichment”. In general, since the data is SNP captured and not shotgun sequenced, I would advocate for the use of the term “genome-wide data” instead of “whole genome data”. Also, I cannot not reconstruct why only 76550 SNPs were genotyped, when (according to lines 247 to 248) “a total of 357,145 targeted SNPs on chromosomes [were] covered by at least one sequence”. Additionally, I would like to know the number of SNPs available for PCA. In Table S2, only the 1240k SNPs are reported, yet the PCA was run on Human Origins data, which is only a subset of the 1240k SNP set.”

Response:

We rephrased the sentence starting on line 416 of the original manuscript to improve clarity. We also added additional details of the Principal Component Analysis in the following sentence. 

“In lines 432 to 436, the authors write “A plausible scenario is that the KYT individual has much of their ancestry from a Proto-Yamna population (Chintalapati et al. 2022), while also being admixed with Serbia/Iron Gates Mesolithic, the latter possibly coming from the Neolithic populations of the Dnipro Valley that have been shown to carry WHG admixture (Mathieson et al. 2018)”. The paper could highly profit from formally testing this hypothesis, e.g. using qpAdm. While PCA and F4-statistics clearly suggest a major proportion of Steppe ancestry in the KYT-SSX individual, qpAdm testing would provide a (citable) percentage and could be furthermore applied to identify the source population that provided the smaller ancestry proportion (which causes the shift away from Yamnaya in the PCA).”

Response:

Regarding qpAdm, with the data available with this publication we don’t have a good model for this individual’s ancestry. We hope to provide more precise estimates in future publications.

The typos noted by Reviewer #2 have been corrected. 

Below please find our response addressing the journal requirements raised in the Decision Letter. 

1. We formatted the manuscript according to the style requirements in the provided style templates.

2. A copy of the completed questionnaire on inclusivity in global research is uploaded as Supplementary Information. We also added the “Inclusivity in global research” subsection to the Methods section that includes the suggested sentence. 

3. All relevant information about the specimens in the study, including all applicable permits, is provided in the Materials and Methods section. We added a statement about permits and regulations to the text (lines 181-185).

4. Repository information will be provided at manuscript’s acceptance.

5. The map in Fig. 1 was made with Natural Earth, naturalearthdata.com, which is from the list of suggested resources provided in your letter. All versions of Natural Earth raster and vector map data found on their website are in the public domain. Please see https://www.naturalearthdata.com/about/terms-of-use/ for details. 

6. The reference list has been reviewed for completeness and correctness. 

We look forward to hearing from you in the near future.

Sincerely, 

Alexey G. Nikitin, Ph.D.

---

## [Editor Report · Decision Letter 1]

24 Apr 2023

Interactions between Trypillian farmers and North Pontic forager-pastoralists in Eneolithic central Ukraine

PONE-D-22-34359R1

Dear Dr. Nikitin,

We’re pleased to inform you that your manuscript has been judged scientifically suitable for publication and will be formally accepted for publication once it meets all outstanding technical requirements.

Kind regards,

Mateusz Baca

Academic Editor

PLOS ONE

Additional Editor Comments (optional):

The authors satisfactory addressed the main issues raised by the Reviewer 2. They provided the correct number of SNPs from the 1230k dataset and the number of SNPs used for PCA. They also provided a more cautious assessment of the analysis of dietary isotopes. In my opinion, the paper is now ready for publication in PlosONE.
---

## [Editor Report · Acceptance letter]

19 May 2023

PONE-D-22-34359R1 

Interactions between Trypillian farmers and North Pontic forager-pastoralists in Eneolithic central Ukraine 

Dear Dr. Nikitin:

I'm pleased to inform you that your manuscript has been deemed suitable for publication in PLOS ONE. Congratulations! Your manuscript is now with our production department. 

Kind regards, 

on behalf of

Dr. Mateusz Baca 

Academic Editor

PLOS ONE